# Using smartphone-GPS data to quantify human activity in green spaces

**Alessandro Filazzola**[1,2]*, **Garland Xie**[1,3], **Kimberly Barrett**[4], **Andrea Dunn**[4], **Marc T. J. Johnson**[1,5], **James Scott MacIvor**[1,2,3]

**1** Centre for Urban Environments, University of Toronto Mississauga, Mississauga, Ontario, Canada, **2** Apex Resource Management Solutions, Ottawa, Ontario, Canada, **3** Department of Biological Sciences, University of Toronto Scarborough, Toronto, Ontario, Canada, **4** Conservation Halton, Burlington, Ontario, Canada, **5** Department of Biology, University of Toronto Mississauga, Mississauga, Ontario, Canada

* alex.filazzola@utoronto.ca

**Data Availability Statement:** All analyses were conducted in R version 4.1.2. All scripts and source codes are available on a public repository

## Abstract

Cities are growing in density and coverage globally, increasing the value of green spaces for human health and well-being. Understanding the interactions between people and green spaces is also critical for biological conservation and sustainable development. However, quantifying green space use is particularly challenging. We used an activity index of anonymized GPS data from smart devices provided by Mapbox (www.mapbox.com) to characterize human activity in green spaces in the Greater Toronto Area, Canada. The goals of our study were to describe i) a methodological example of how anonymized GPS data could be used for human-nature research and ii) associations between park features and human activity. We describe some of the challenges and solutions with using this activity index, especially in the context of green spaces and biodiversity monitoring. We found the activity index was strongly correlated with visitation records (i.e., park reservations) and that these data are useful to identify high or low-usage areas within green spaces. Parks with a more extensive trail network typically experienced higher visitation rates and a substantial proportion of activity remained on trails. We identified certain land covers that were more frequently associated with human presence, such as rock formations, and find a relationship between human activity and tree composition. Our study demonstrates that anonymized GPS data from smart devices are a powerful tool for spatially quantifying human activity in green spaces. These could help to minimize trade-offs in the management of green spaces for human use and biological conservation will continue to be a significant challenge over the coming decades because of accelerating urbanization coupled with population growth. Importantly, we include a series of recommendations when using activity indexes for managing green spaces that can assist with biomonitoring and supporting sustainable human use.

## Author summary

In urban areas, green spaces represent important places for recreation, preservation of biodiversity, and delivery of ecosystem services, such as managing stormwater and

that can be found at https://github.com/afilazzola/CUERecreationEcology. Data for the characteristics and summarized activity values for each green space tested are also publicly available https://doi.org/10.6084/m9.figshare.21304767.v1.

**Funding:** This research was funded by a Post-Doctoral Fellowship awarded to AF by the Center for Urban Environments and School of Cities at the University of Toronto, Canada. GX was funded by an Ontario Graduate Scholarship, a Center for Environmental Research in the Anthropocene Graduate Fellowship, and NSERC CREATE funding (# 401276521) awarded to JSM. The funders had no role in study design, data collection and analysis, decision to publish, or preparation of the manuscript.

**Competing interests:** The authors have declared that no competing interests exist.

reducing extreme heat. How people use green spaces and their impact on urban biodiversity is not well understood, particularly because it is difficult to monitor human activity. We used anonymized GPS data from smart devices to quantify green space use in Southern Ontario. We found a strong correlation between our estimates of mobile device activity and green space visitation rates determined from reservation data. We also found that users often spent most of their time on trails and that there were correlations between human activity and tree composition. We provide one of the first analyses exploring how people use urban green spaces using GPS data and the potential link to urban biodiversity.

## Introduction

Cities are rapidly expanding, creating new challenges for managing green spaces. More than half of the global population currently lives in cities and that number is projected to increase to almost 90% by the end of the century [1,2]. As cities increase in size and area, green spaces including remnant natural areas, protected reserves, and urban parks, face growing stressors from human activity. Direct human use of green spaces can negatively impact urban wildlife including trampling, litter, the introduction of non-native species, and pollution [3–6]. However, both managed and unmanaged green spaces are important for city residents as a place for exercise, recreation, socialization, and supporting mental well-being [7–10]. This need has been amplified during the COVID-19 pandemic as people have increasingly sought out green spaces when indoor areas were closed or with increased risk of infection [11–13]. Thus, managing green spaces is a delicate balancing act between utility for people and the conservation of biodiversity.

One of the main limitations in effectively managing green spaces is the uncertainty around how and when people use these areas. Some managed parks use a reservation-based system with controlled points of entry, whereas other green spaces have multiple unrestricted access points. Trails are created to facilitate human movement and reduce disturbance to biodiversity, but visitors will still venture off-trail or erode new paths of easily navigable terrain [14]. Determining areas of high disturbance (i.e., high traffic), potential off-trail use, and overlap with sensitive species, can all be achieved through understanding human activity in green spaces. However, capturing human activity at a resolution fine enough for management, such as less than 100 x 100 m, is challenging. Typical methods for quantifying human activity include record-keeping visitors at entrance points, video monitoring, or *post-hoc* assessment of visitor impacts, such as campsite use [15–17]. Unfortunately, these data often neglect any spatial component of what visitors do outside of control points. Visually tracking visitors as they move within green spaces can be both cost-prohibitive and potentially intrusive. Using social media can be effective to track actions and activity from geotags of publicly shared images, posts, or tweets [18,19], but these data can be biased towards individual behaviours and be biased towards intentional points of interest [20]. With the widespread adoption of mobile smartphones and other smart devices, using anonymized GPS data can be an effective tool in determining patterns in the use of green spaces.

Connecting biodiversity observations to smart device activity data can pose a unique set of challenges beyond validating human activity patterns. Any correlation between species and human activity could occur because of multiple pathways including i) the species is relatively resilient to human disturbance and thus persists when others cannot, ii) the species or species' habitat is attractive to visitors, iii) the property coincidentally is dominated by this species and is very accessible (e.g., walking distance to residential areas), or iv) any combination of these

three factors. Teasing apart which of these pathways is relevant can be challenging because correlations between human activity and biodiversity may be because of aesthetic appeal or accessibility. While GPS data from smart devices have broad spatial and temporal coverage across a region, biodiversity data is often restricted to long-term monitoring plots that are static in location or multiple experimental sites that are short-term [21]. Biodiversity data are also rarely collected daily or cover a broad spatial area, presenting a challenge when trying to connect these two disparate types of data (e.g., evaluating the effects of human activity on biodiversity). Additionally, biodiversity surveys are often conducted away from areas with high human activity (e.g., trails, playgrounds, picnic areas) in more naturalized areas, reducing the chance that any overlapping human activity would be recorded. Using community science (e.g., iNaturalist, e-bird, Bumble Bee Watch) can be an effective tool for obtaining surveys with broad spatial and temporal coverage of green spaces [22,23], but these types of data are inherently correlated with smartphone use because of the mobile applications they require. A preliminary exploration of biodiversity and GPS data from smart devices would include examining the relative use of land cover types in green spaces to determine if certain areas, particularly where there is sensitive habitat, receive disproportionate levels of human activity.

Management of urban green spaces can be complex trying to balance different property types, land covers, and public uses. We partnered with Conservation Halton (www. conservationhalton.ca), a local conservation authority within the Conservation Ontario (www. conservationontario.ca) network responsible for natural areas, protected reserves, and urban parks in the regional municipality of Halton. Conservation Halton is responsible for managing a watershed that spans 1000 km$^2$ of land containing different ecosystems, including forests (106 hectares; 7.5% coverage), riparian vegetation (7 km in total length; 14.5% coverage), and grasslands (130 hectares; 10.9% coverage) [24]. To improve ecosystem service delivery, Conservation Halton is working to increase natural land coverage above a particular threshold but must also balance naturalization with the provisioning of recreational opportunities [24]. Maintaining this balance is challenging since there are over 1.2 million visitors annually to the Conservation Areas due to their proximity to large urban centers (e.g., Hamilton, Milton, Mississauga, and Oakville) [25] with many of these municipalities being the fastest-growing cities in Canada [26]. Common social features of this landscape are recreational activities such as hiking, dog walking, cross-country skiing, and picnicking. Management of the 53 diverse properties within Conservation Halton's jurisdiction represents some of the common challenges associated with land managers responsible for urban green spaces.

GPS data from smart devices can be a powerful tool in managing green spaces, but methods are needed that can properly quantify patterns of human activity. The purpose of our study was to describe how an activity index of anonymized GPS data from smart devices could be used for human-nature research, particularly looking at the associations between park features and human activity. GPS data from smartphones has been used previously to estimate trail use, green space access, and outdoor recreation patterns [13,27,28], but many studies rely on volunteer participants representing a fraction of green space users. Using Mapbox Movement (www.mapbox.com/movement-data) we obtained an anonymized activity index representing human density aggregated to 100 x 100 m grid cells and two-hour windows. In the following study, we developed methods for the synthesis, management, and analysis of anonymized GPS data from smart devices in Conservation Halton green spaces by answering the following three questions:

1. How does the anonymized activity index (comprising both activity density and activity coverage) compare to traditional measures of human activity in urban green spaces, such as reservation data or trail density?

2. What are the challenges associated with using the activity index to infer human presence in green spaces, particularly for land managers?

3. Can the activity index be used to correlate patterns of human activity to landscape features and tree composition?

## Results

### Patterns of human activity

The adjusted activity density was found to accurately capture human visitation within green spaces. All green spaces had at least one grid cell with activity values above the threshold used to anonymize the data. Green spaces had on average 40.6% (SE ± 3.3%) of the total area with detectable GPS-location data (Table 1). Reserve areas had the lowest percentage of activity coverage (Table 1) as would be expected for lands where access is limited. For the green spaces where reservations were required, we found a strong positive relationship between the total number of reservations and the adjusted activity density ($F_{1,8} = 10.7$, $p = 0.011$, $R^2 = 0.63$; Fig 1). The relationship between the number of reservations and (adjusted) activity density was mediated by day-of-week ($F_{1,8} = 6.95$, $p = 0.029$), where on average, reservations across green spaces was 2.5% higher on weekends despite the weekend representing fewer days. This pattern suggests that for the same number of reservations, people often spend more time at these green spaces on the weekend relative to weekdays.

### Unique metrics of human activity

The activity index can provide greater spatial and temporal resolution for human activity relative to tracking visitation patterns through park entrances. Many green spaces had hot spots of activity values on trails, whereas adjacent areas (*i.e.*, 'off trail') had substantially lower activity values. For example, two green spaces (Hilton Falls and Kelso Conservation Areas) had high activity patterns within their trail network (Fig 2), and on average, parks with higher trail densities were found to have higher amounts of activity density ($F_{1,16} = 8.60$, $p = 0.0097$, $R^2 = 0.35$; Fig 3A). Similarly, green spaces with high densities of trails also correlated with more area of activity coverage, *i.e.*, areas of the green space containing any human activity ($F_{1,9} = 6.57$, $p = 0.035$, $R^2 = 0.36$; Fig 3B). The percent of activity on-trail was significantly correlated with activity coverage ($F_{1,9} = 18.0$, $p = 0.002$, $R^2 = 0.63$; Fig 3C), suggesting increased green space use typically occurs on trails.

### Activity patterns and the environment

Activity density varied considerably by the land cover classes described by Conservation Halton (ELC-CC, Ecological Land Classifications Community Classes). Forest and cultural land

**Table 1. General characteristics of the green spaces within Conservation Halton's jurisdiction including land type, whether properties are actively managed, number of properties, and average property size.** Activity coverage represents the percentage of an area within the green space that has any human activity determined from the Mapbox data. A list of all green spaces and characteristics can be found in S2 Table.

| Land Type | Managed | Properties | Property Size (km²) | Activity coverage (%) |
|---|---|---|---|---|
| Conservation Area | Managed Land | 7 | 3.68 | 59.8 |
| Conservation Area | Non-Managed Land | 8 | 0.88 | 48.4 |
| Natural Area | Non-Managed Land | 16 | 0.5 | 44.5 |
| Other | Non-Managed Land | 9 | 0.1 | 34.3 |
| Reserve Area | Non-Managed Land | 13 | 0.24 | 26.3 |

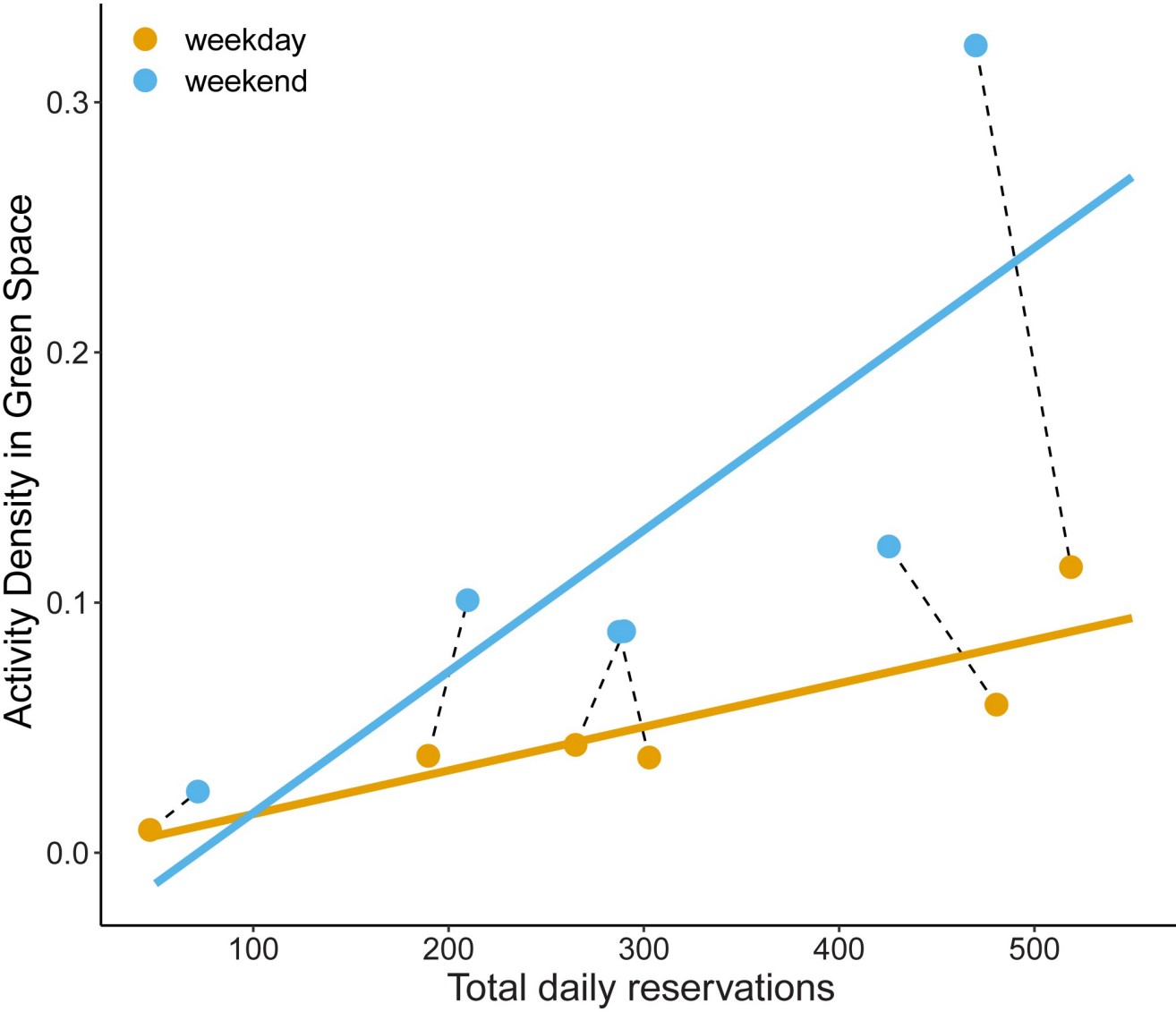

**Fig 1. The total daily reservations for Conservation Halton green spaces that had reservation-only entrances were strongly positively associated with activity density (mean effect ± SE = 0.00017 ± 0.00013).** On average, activity density was much higher on weekends compared to weekdays (mean effect ± SE = 0.00039 ± 0.0002). Dashed lines connect the same green space for weekends and weekdays. Reservation and activity density represent the totals within each conservation authority property between June and August 2020.

classes were associated with the highest proportion of grid cells with human activity, followed by talus and cliff (Fig 4A). However, relative to the abundance of the land classes in each property, rock formations, including talus, cliff, crevice and cave, and bluff, were disproportionately visited (~ 75%) relative to other land types (Fig 4B). By contrast, forests and cultural land classes had a lower proportion of activity density relative (<50%) to their abundance among green spaces.

We found that our measures of human activity were significantly correlated to patterns of tree composition across the sixteen green space properties ($F_{15}$ = 2.74, p = 0.001; Fig 5), explaining 25% of the variation in species composition. Although few species were uniquely identified to correlate with human activity, there were some correlations observed. Yellow birch (Betulaceae: *Betula alleghaniensis*) and black ash (Oleaceae: *Fraxinus nigra*) were both

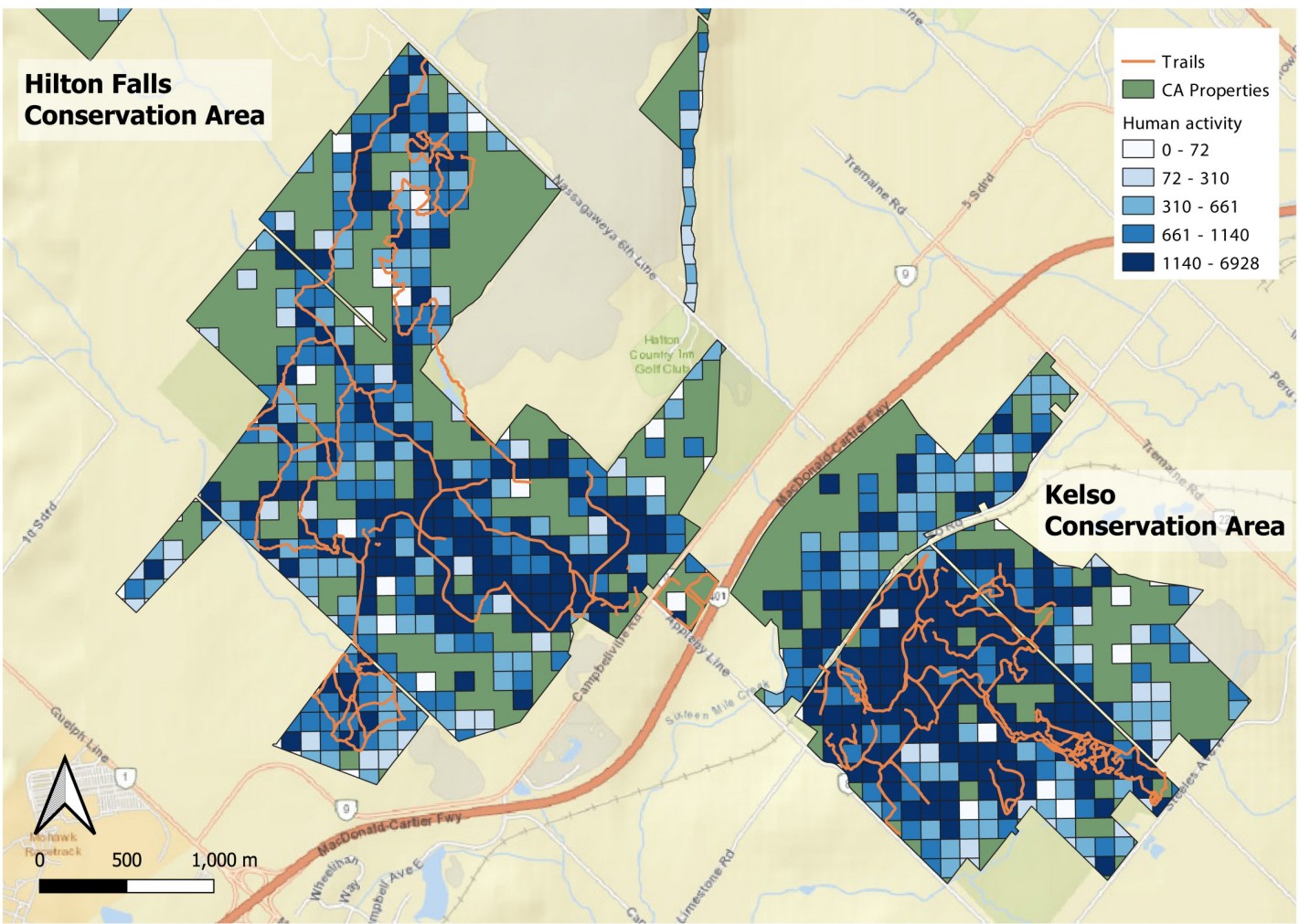

**Fig 2. A representation of anonymized activity density in two Conservation Halton green spaces, Hilton Falls and Kelso Conservation Areas.** Each blue grid-cell represents a 100 x 100 m pixel of activity density with darker colours representing higher densities of human activity. Values represent the maximum activity density value observed across all hourly timeframes for the three-month time-series. Orange lines within property boundaries are official trails managed within the green space. Green areas within property boundaries have activity levels too low to be available in the Mapbox dataset, potentially representing refugia in the park where human activity is negligible. Alternately, some of these low-activity areas are small water bodies or inaccessible areas because of terrain. Maps were created using Open Street Maps (https://www.openstreetmap.org/).

positively correlated with activity density and proportion of weekend activity (Fig 5). Some species appear to be negatively associated with activity density including bitternut hickory (Juglandaceae: *Carya cordiformis*), ironwood (Betulaceae: *Ostrya virginiana*), and American elm (Ulmaceae: *Ulmus americana*). However, these three species are also relatively uncommon compared to the other tree species examined.

## Discussion

Our study represents one of the first that examines relationships between human activity and green space using anonymized GPS data from smart devices. We found a significant

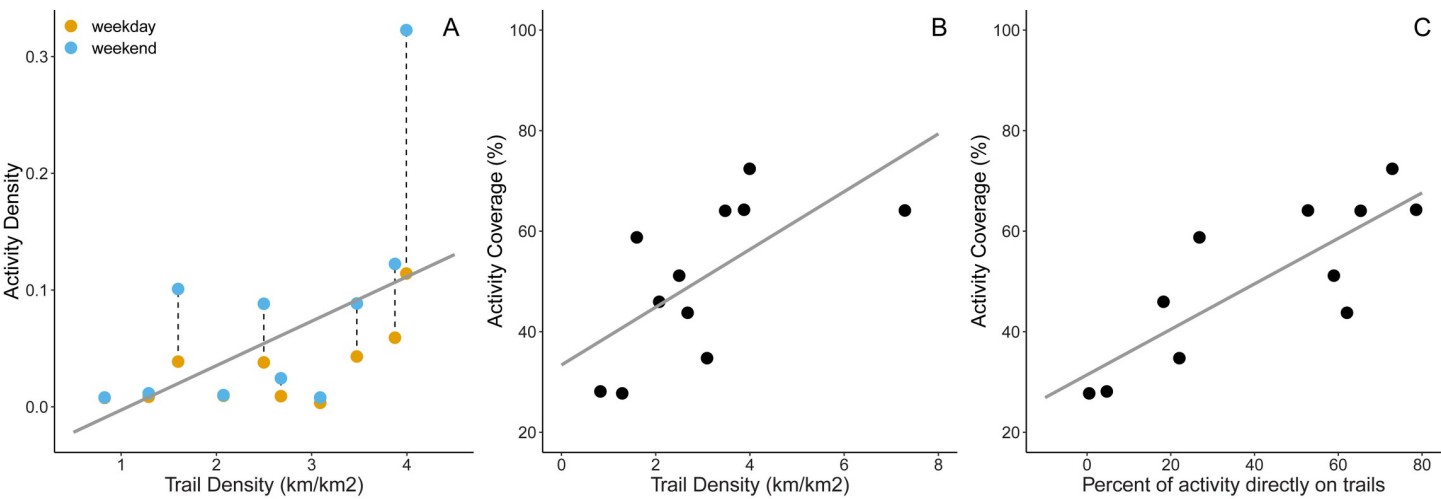

**Fig 3. Patterns of trail use with activity density and coverage.** Green spaces with higher trail densities were found to have significantly higher activity patterns (mean effect ± SE = 0.021 ± 0.018; Panel A). Dashed lines connect the same green space for weekends and weekdays. There was also a positive relationship between the area of activity density with trail densities (mean effect ± SE = 0.057 ± 0.022; Panel B), where a higher proportion of trails result in more coverage of the green space with human activity. The percentage of activity on trails (*i.e.*, on-trail vs. off-trail use) was positively correlated with activity coverage (mean effect ± SE = 1.47 ± 0.35; Panel C).

correlation between the number of visitors (based on reservation data) and activity density ($R^2$ = 0.63; Fig 1), demonstrating that smart device activity effectively captures human activity in green spaces. However, it is important to note that pre-processing is required to reflect the activity more accurately in green spaces. Both activity density and coverage proved effective at capturing human activity patterns in green spaces including patterns of trail and land cover classes (Figs 3 and 4). For land managers looking to balance human use with biological conservation in green spaces, we illustrate that these data are both powerful and accessible for pinpointing hot spots of human activity, prospective ecological refugia (i.e., where human activity is low), and encroachment of activity on restricted areas. This information can be used to *a priori* plan biomonitoring to capture impacts along a gradient of human activity level.

## Description of observed patterns and interpretation

Activity patterns varied considerably, but predictably among green spaces. Off-trail use is a significant problem in conserving biodiversity, causing disturbance, trampling, and introduction of non-native species [6,29,30]. In the green spaces evaluated, we found that most visitors appear to remain within the designated use areas, with the highest activity observed along trails (Figs 2 and 3) or in recreational spaces (e.g., picnicking areas) (Fig 4). Still, the activity outside of designated use areas persisted across green spaces with the percent of activity on-trails dropping below 5% of total activity (e.g., Kilbride). Some of these green spaces have unofficial trails that are managed by non-profits or local communities. For example, the Bruce Trail Conservancy manages a 904 km trail that intersects some of these green spaces but that is independent of Conservation Halton (www.brucetrail.org/). Future land managers interested in relating trail networks to human activity may need to aggregate trail locations from multiple data sources, such as AllTrails (www.alltrails.com/) or TrailForks (www.trailforks.com/). The activity data used here can also guide managers to areas of frequent or abundant human presence but where no trails exist, to determine where off-trail incursions are most common. For green spaces in Conservation Halton, the highest activity of off-trail use appeared near the entrances or in areas between adjacent trails (e.g., Fig 2). These perceived negative impacts to green spaces could be flipped to a positive if human behaviour (*i.e.*, where people go off-trail the

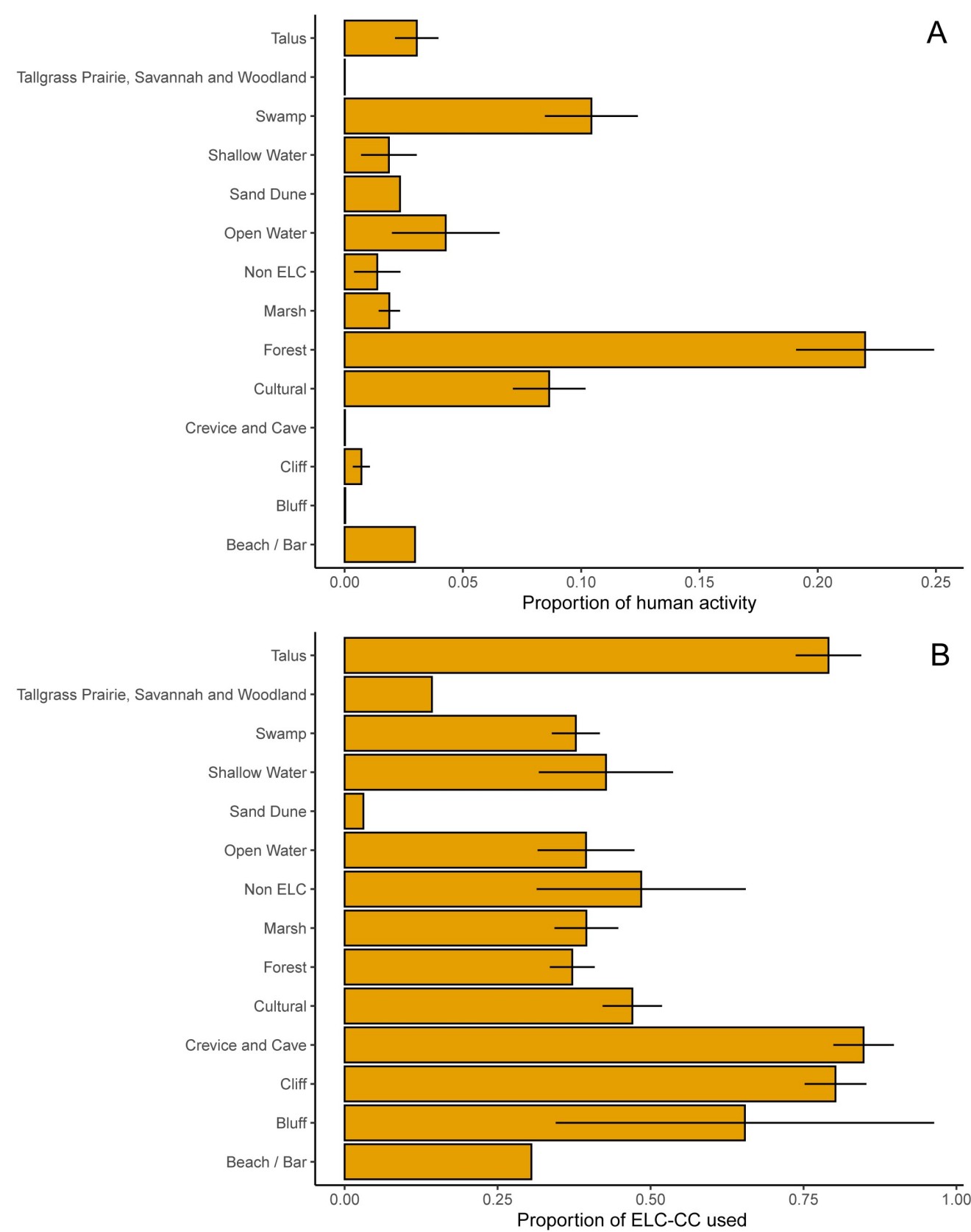

**Fig 4. Patterns of human activity in each of the Ecological Land Classifications Community Classes (ELC-CC) for Conservation Halton green spaces.** We calculated the proportion of human activity by using any grid cell that had activity throughout the 3-month timeframe relative to the land cover of that grid cell. *The proportion of human activity* by ELC-CC represents the percent of human activity within the property separated by each ELC-CC (Panel A). *The proportion of ELC-CC used* represents the percent of human activity in that ELC-CC relative to the total coverage of that class within the property (Panel B).

most) is useful for guiding future trails and accessibility. However, biodiversity in these areas will be more susceptible to disturbance and if off-trail activity occurs in sensitive habitats, this human activity could be monitored and modified. Additionally, while trails are typically managed to mitigate impacts to the landscape, many are still constructed in ecologically sensitive areas [e.g., 31]. Thus, high activity on trails are not enough information alone to determine effective management.

## Limitations of location data

Anonymized GPS data from smart devices is a powerful tool with broad spatial and temporal resolution, but there are biases and considerations for its use (Table 2). The challenges we encountered when estimating human activity can be generalized into social, technical, and data issues (Table 2). Although mobile device use has expanded rapidly across the globe [32], there remain large differences among regions and demographics [e.g., 33,34]. In areas where mobile device adoption is high, smart device locations may more accurately reflect human activity relative to other location-based data (*e.g.*, social media, geotagged photos, *iNaturalist*). Our study took place in Canada where LTE mobile networks cover 99% of the population [35] and 80% of Canadians report having a mobile data plan for personal use (www150.statcan.gc.ca). However, even within countries, there are differences in mobile device use between rural, suburban, and urban communities. In the United States, rural Americans have consistently fewer mobile devices relative to residents of urban or suburban areas [36]. In Canada, 81% living within a city metropolitan area (CMA) had mobile data plans compared to 73% in non-CMA areas (www150.statcan.gc.ca). The devices and the software applications used will also be prone to biases (Table 2). There can be variations in quantifying activity caused by different accuracies among devices, operating systems, software, and location [37]. For instance, smart devices can vary between 5–10 meters in GPS positioning based on hardware [27,38]. The choice of a software application by the device user can also determine activity patterns. For example, a person using a ride-sharing application is more likely to have location services turned on, whereas a person in a green space may not have any application open. As green spaces are often viewed as a place to "disconnect" or be engaged in activities that discourage mobile device use (e.g., swimming, jogging), activity patterns may be less accurate than when compared to roads. These biases are important when considering the expansion of the applicability of human activity data to other demographics or regions (Table 2).

Some of the biases associated with using smart device data to infer human activity have been identified in previous studies, but there are specific considerations concerning green spaces. For example, the activity index was anonymized by aggregating activity patterns to coarse resolutions to prevent harassment, crime, or injustice [39–41]. This method facilitates anonymity and prevents tracking individual behaviours, activity by demographics, or the fine-scale resolution of activity patterns (e.g., < 100 m). However, most green spaces found in urban areas are small, so discerning activity within the green space relative to nearby cities within these aggregated areas can be difficult. This becomes particularly problematic on green space boundaries that are delineated by private residential properties or high-traffic roads. Smart device data are rarely separated by mode of transportation (*e.g.*, pedestrian, cyclist, motorist) and thus differentiating between cars driving along the boundaries and hikers within

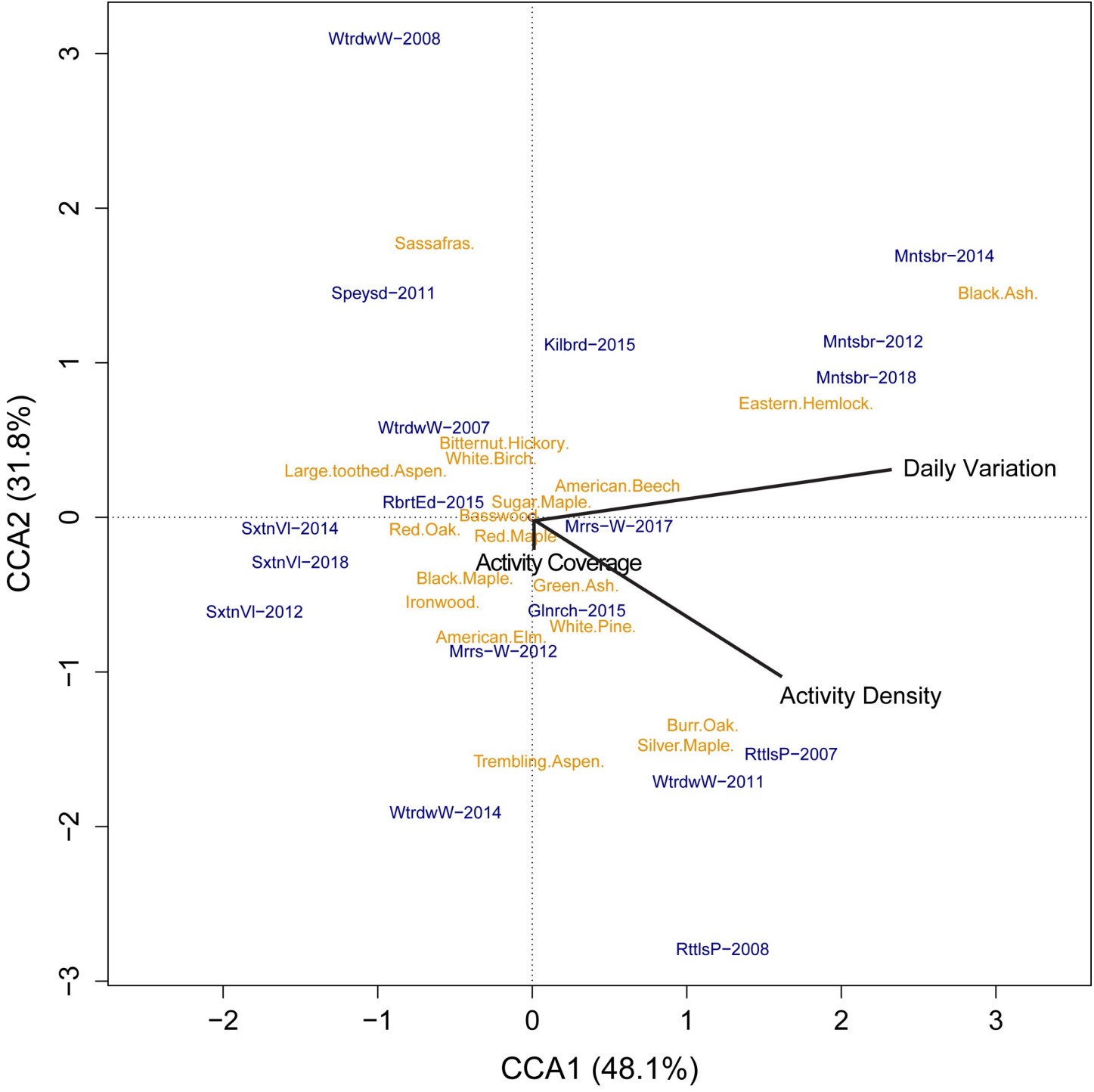

**Fig 5. Correlations between human activity and tree composition in green spaces ($F_{1,15}$ = 2.74, p = 0.001).** The partially constrained correspondence analysis (pCCA) explained 79% of the variation in tree species among green spaces with an adjusted $R^2$ = 0.17. Names in orange represent the abbreviated common names of tree species found at each site (blue). Black represents the constrained predictors of activity density, proportion of weekend activity, and activity coverage (percent coverage of green space). Year was included as a conditioning matrix to partial out any interannual variability.

**Table 2. Considerations for future users of anonymized location data for estimating human activity.**

| Issue | Key Studies | Challenge | Details | Examples |
|---|---|---|---|---|
| *Social* | [39–42] | Privacy | Data should not be too specific because it can infringe on people's privacy or be used for nefarious purposes. | Data are anonymized and aggregated to a coarse spatial resolution. |
| | | Ownership differences | The number of mobile devices differs among the demographics of people. | More affluent communities will likely possess more devices and thus have higher activity patterns. |
| | | Safety | The use of devices may vary depending on location or security. | Out of safety concerns, individuals may avoid using their devices. |
| | | Behaviour of individual | Use of the device may vary depending on the activity of individual | When exercising someone may not use their device. |
| | | Behaviour on phone | Location tracking services may vary depending on application use on a smart device. | Ride-sharing applications will more frequently report location relative to a messaging application. |
| | | Land type | Certain land types are inherently more likely to have higher activity. | Roads are vectors for transportation and thus will typically have high activity relative to other areas. |
| *Technical* | [27,37,38] | Accuracy | Not all devices or networks equally estimate the location of a device. | Older cell phones may have a larger radius of location accuracy. |
| | | Coverage | Some areas have limited or no coverage of a cellular network because of geographic barriers or the placement of towers. | Rural green spaces may have low coverage because of topography preventing coverage and infrequent towers. |
| | | Noise | To minimize privacy concerns, activity data are often masked below a certain threshold or have a random amount of activity added. | Areas with low human activity will have no values reported. |
| | | Time | Over time, the number or usage of devices can increase. | Devices with location services were less common 10 years ago than today. |
| *Data* | [43,44] | Training | Requires expertise in spatial analysis and high-performance computing. | Activity data are often vectored tiles based on spatial extents. |
| | | Cost | Data are not publicly available and require purchase from companies. | Activity data is costly even when purchased as snapshots based on certain timeframes or areas. |
| | | Management | The size and type of data often require high-performance computing resources. | Regional datasets often exceed the storage and memory capabilities of personal computers. |

the adjacent green space can be difficult at coarse scales. Similarly, determining the behaviour of individuals can require some assumptions (Table 2). One can infer activity at a beach or picnic area could represent swimming and socializing respectively, but neither is definitive nor requires knowledge of how land cover matches the activity type. Another common challenge with anonymized activity data is thresholding to remove activity patterns below a certain level to prevent tracking select individuals on private properties. Green spaces often have relatively less activity compared to adjacent paved city spaces (e.g., roads, buildings), causing some areas to report no activity when activity is low. One approach to resolving the above challenges is to examine every green space case-by-case to validate activity patterns. However, for municipal land managers responsible for many properties, this approach is laborious and subjective. Our approach of removing grid-cells based on high overnight activity is reproducible and systematic but can lower the number of available grid cells for analysis.

## Human activity and the environment

Certain land cover types had higher activity relative to others. Forests were the most visited land cover, but also represented the largest area of the green spaces, particularly surrounding the trail network (Fig 4). Cliff and rock formations were visited disproportionately high relative to their cover, likely because these areas have scenic points of interest (e.g., lookouts, waterfalls) and a developed trail network. Typically, these land cover types would also be attractive for rock climbing activities (i.e., bouldering, rappelling, climbing), but these activities were prohibited during the 2020 season because of the pandemic. Rock formations are

consistently popular attractions in green spaces [e.g., 45], but can pose a hazard both to human health from climbing accidents and to local biodiversity from trampling [46,47]. Many green space visitors seek quiet, uncongested areas [48,49], and overcrowding may drive negative experiences among users, which decreases support for conservation [50]. Yet, some areas in green spaces will inherently be more sensitive than others. Knowledge about preferred land cover types can inform land managers to either prioritize conservation potential in areas with low human activity (e.g., swamps, marshes, or grasslands) or implement measures (e.g., fencing) to mitigate impacts to high-activity areas.

Associating human activity with patterns of biodiversity is inherently challenging. We demonstrated preliminary evidence that human activity interpreted from smart devices is related to patterns of tree composition in green spaces (Fig 5). However, it is important to note we are not implying causation or mechanistic relationships and instead are providing a heuristic of the potential for activity data to be related to green space management. For instance, we saw a negative relationship between human activity and ironwood, *Ostrya virginiana* (Betulaceae), but this species is more commonly found in swamps or high-moisture areas [51], within which we saw markedly lower human activity. Conversely, yellow birch, *Betula alleghaniensis* (Betulaceae), has a visually appealing bark and thus may attract greater human activity. Additionally, trails are often designed to direct people to certain habitat types because of accessibility, ease of construction, or ease for visitors to travel. Eastern hemlock, *Tsuga canadensis* (Pinaceae), is relatively tolerant to disturbance compared to other species, and thus may be more likely associated with higher intensity of human activity [52]. There may also be indirect factors driving the observed relationships, such as high white-tailed deer densities (*Odocoileus virginianus*), which have a significant effect on tree composition in Southern Ontario [53,54] but may also be appealing to visitors. Other types of biodiversity data may be useful for disentangling some of these patterns. We used long-term monitoring data of trees that is relatively robust to survey biases but is restricted to select survey locations and certain years. Community science databases (*e.g.*, *iNaturalist*, *eBird*) would have larger spatial and temporal coverage allowing for greater overlap with the activity data, although most will be inherently correlated to activity data since they have smartphone applications. Relating patterns of biodiversity to smart device activity is thus a promising opportunity that requires careful consideration of data sources and experimental design.

## Implications

The prospects of anonymized GPS data from smart devices for managers of green spaces and more broadly, to balance human impacts with the engagement of the environment, is exciting. We show it surpasses information available from counting visitors at the entrance by including human activity hot spots and cold spots, off-trail movement, and human behaviour. This information is crucial to support the connection of visitors to important park features that deepen care or appreciation of the environment. Land managers can also improve the accessibility of the green space and develop more parsimonious trail routes that mitigate off-trail activity. Anonymized activity data is also helpful for quantifying visitation rates and patterns in ungated or unstaffed green spaces with no reservation system or tracking mechanism in place. With an improved understanding of human activity across green spaces, biomonitoring inventories can be coordinated in ways that capture biodiversity information across gradients of human activity levels. This could be critical to disentangle local disturbance and modification to community composition, loss of sensitive species, and management of at-risk species. Since human activity in green spaces is correlated with invasive species propagules [3], biomonitoring inventories designed to include sites along human activity gradients may be critical to

proactively manage invasive species and mitigate economic costs associated with eliminating established species. Location data from smart devices may thus provide a stronger proxy of human activity, and consequently propagule pressure, relative to landscape analyses such as distance to roads.

The activity data collected by Mapbox is recorded hourly and so continuous monitoring of green spaces for changes in human behaviour is possible. With the COVID-19 pandemic, many local parks have seen a significant increase in the number of people visiting, and with this comes a wider range of behaviours [11,12]. For instance, Conservation Halton saw an increase in visitors to approximately 1.2 million visitors over 10 months in 2020 (many green spaces were closed in April and May because of the pandemic) relative to only 1.1 million visitors in 2019 for all twelve months. Knowledge of when green spaces are visited and wherein the park visitation is highest will permit refinement of both biomonitoring, management, and engagement to more effectively (and economically) link the needs of humans to access nature and recreation opportunities while also conserving biodiversity in green spaces.

## Data and methods

### Mapbox movement data

We obtained anonymized GPS data from smart devices from Mapbox for the Greater Toronto Region (43.23˚ N– 44.35˚ N, 78.83˚ W—80.26˚ W) which includes the Regional Municipality of Halton and its surrounding areas. Mapbox is a private company that specializes in location data with products for application development. The data they provide is referred to as an "activity index" representing a density of smart devices within a 100 x 100 m grid-cell. The methods Mapbox uses to create the activity index can be broken down into four steps: 1) Collection, 2) Calibration, 3) Anonymization, and 4) Normalization. The collection includes location data from any smart device that uses the Mapbox software development kit (SDK). Location data can come from the GPS within the device, cell tower triangulation, or router indexation if the device connects to a wireless network. The collection pipeline aggregates repeated location information from a device into five-minute intervals within a grid-cell, but discards the start and end of a trip to assist in anonymity. Therefore, a high activity index within a grid-cell can be both from a high density of smart devices or continuous movement of the same smart devices. The calibration pipeline adjusts for fluctuations in the volume of telemetry data during collection. Because the location data comes from thousands of smart device applications, each with their use patterns, the team at Mapbox needs to apply adjustments to ensure the data reflect real-world conditions. The calibration steps are extensive and a discussion of the methods involved can be found here (https://www.mapbox.com/blog/calibrating-mapbox-movement).

Devices are aggregated to these grid-cells for anonymity by preventing the tracking of individuals or identifying human patterns within private areas. To further anonymize the activity index, Mapbox removes activity levels below a threshold and adds an unspecified amount of noise to prevent re-identification [e.g., 41,55]. The applied threshold and added noise are generated using a machine-learning algorithm developed by Mapbox. Lastly, Mapbox normalizes the activity index within a given country (i.e., Canada) and scaled each time interval to a baseline of the mean activity patterns for January of the respective year. The activity index is thus unitless without any real-world equivalent, such as the density of people, number of smart devices, or time spent in an area. Mapbox recommends that the activity index be only used for relative comparisons in human activity rather than any analysis of a specific data point or grid-cell. Additionally, for the above reasons, Mapbox emphasizes that comparisons made over short time scales (e.g., within a year) and similar regions (e.g., within the same country) will be

more relatable than those across larger time-periods and other regions. For more details on how Mapbox calculates its proprietary activity index, see their online guide (https://docs.mapbox.com/data/movement/guides/).

The activity index was provided by Mapbox in 100 x 100 m grid-cells across Halton Region for June, July, and August 2020 (S2 Table). Each grid cell has a monthly average value for 2-hour time windows throughout a complete 24-hour day. Monthly averages are also separated into weekdays (Monday-Friday) and weekends (Saturday and Sunday). We found the intersection of each grid cell with the green spaces managed by Conservation Halton through an iterative loop (function *st_intersection*; package *sf*). Grid cells that were found to intersect on Conservation Halton properties often ended up masked to an area smaller than the full 100 x 100 m grid-cell (see Fig 2).

A significant challenge with using the smart device activity for green spaces was the accidental inclusion of activity outside of the green space. Roads and highways were especially challenging when adjacent to property boundaries, causing high activity patterns that are likely not reflective of the activity within the property. Removing grid cells individually based on proximity to roads is labourious, requires spatial information about roads, and can be subjective. For a more systematic approach, we identified any grid cell with human activity between 12–6 am. Many of the green spaces are closed to access overnight and the remaining properties likely experience substantially lower traffic compared to daytime hours. Additionally, mobile device use is typically lower during these hours [56]. The activity in these areas between 12–6 am is therefore likely below the threshold identified by Mapbox for human activity. Conversely, roads and adjacent commercial operations remain active during overnight hours. Therefore, we excluded any grid cell with activity during these select hours to remove activity outside of the green spaces from being reported. We validated this approach by examining the association of the activity index and roads in all green spaces. As expected, roads and especially busy highways had the most night-time activity, whereas the interior of green spaces was almost exclusively without any activity. See S1 Appendix for a discussion of this exclusion based on high night-time activity.

## Green space data

As a case study for using anonymized GPS data from smart devices with green spaces, we selected 53 green spaces managed by Conservation Halton in Ontario, Canada. As part of the Conservation Ontario (https://conservationontario.ca) network of conservation authorities, Conservation Halton is a conservation authority empowered by the provincial government to manage green spaces for biological conservation, the preservation of ecosystem services, and human recreation. The 53 green spaces encompass a range of management types including conservation areas used for recreation and conservation, natural areas where human visitation is not facilitated (i.e., no parking lots or trails), reserve areas where human activity is limited (e.g., fencing), and other areas that include stormwater diversion channels and urban parks (Tables 2 and S2). While there were 53 separate green spaces, not all had data available for our analyses. For our subsequent analyses, five sites had reservation data, eleven had official trail networks, eleven had tree diversity measures, and all had land classification information.

During the summer of 2020, visitation to seven of the green spaces managed as conservation areas was controlled through reservations because of the Covid-19 pandemic. Individuals with reservations were allowed to visit the green space between 9 am and 6 pm for a maximum of 2 hours. These seven green spaces are among the most popular within Conservation Halton with features including waterbodies, rock formations, look-outs, and well-developed trail networks. We obtained the reservation data for visitors that attended these seven properties for

June, July, and August 2020. Each reservation included the number of individuals, the time of check-in, and the park visited. Additionally, we obtained information about the property boundaries, Ecological Land Classification, and officially managed trail network from Conservation Halton's open data portal (www.conservationhalton.ca/mapping-and-data). The Ecological Land Classification categorizes land formations and vegetation communities to assist in the characterization of the landscape [57]. Through ground surveys, lands are grouped into different classifications such as marsh, forest, dune, and swamp.

To test the relationship between human activity and local landscape features we compared land cover and tree composition in different green spaces. Land cover was obtained from an Ecological Land Classification that was conducted by Conservation Halton. Ecological Land Classification uses a systematic approach to convert high-resolution aerial imagery into digitized spatial polygons assigned a Community Class [57,58]. Examples of the Ecological Land Classification Community Classes (ELC-CC) include forest, cultural (e.g., recreational areas), talus (i.e., scree), and open water. Tree surveys were conducted at eleven unique sites in Conservation Halton's watershed jurisdiction between 2006 and 2019. Terrestrial vegetation monitoring protocols established by the Ecological Monitoring and Assessment Network were followed [59]. Within permanent sample plots, each tree with a diameter at breast height (dbh) of 10 cm or greater was individually tagged and basal area was calculated over subsequent visits. Between 10–20 permanent plots were set up in each of the eleven sites and we averaged across all plots to estimate the mean basal area per species in each green space and survey year. Although the activity data was acquired for 2020, the composition of tree species at these sites remains relatively constant between years. To confirm composition did not change significantly between years, we conducted a constrained correspondence analysis with year as a predictor and found it did not significantly affect tree species ($F_{1,26} = 1.26$, $p = 0.23$).

## Data analysis

We used two metrics of human activity based on the Mapbox activity index: activity density and activity coverage. Activity density (Eq 1) was calculated by taking the grid-cells determined to not have high overnight activity and multiplying the activity value by the area of the grid-cell. When the green space polygons intersected with the Mapbox polygons, many were reduced to areas that were only a fraction of the full size, but the activity index for that grid cell represents the entire grid-cell. Therefore, we multiplied by the area to scale the activity based on the area occupied. To quantify activity density for an entire green space, we used the sum, rather than median or mean, because the number of grid cells varies over time due to the thresholding of activity that is applied to anonymize the data (i.e., no grid cells have zero values, they are simply absent). Thus, we divided the sum of adjusted activity across all grid-cells within the green space and across all three months by the area of that green space. Since the provided Mapbox index represents a normalized proportion, our adjusted metric of activity density relates to the proportion of smart device activity in a grid-cell and a given 2-hour window. The activity index is thus weighted both by the number of devices and the duration the devices spent in the grid-cell of interest.

We also calculated activity coverage for every property (Eq 2). If any grid-cell had an activity index value for any of the time-periods within our dataset, that grid-cell was treated as having human activity. We totaled the area identified with human activity and divided it by the total area of the property to determine the percent area (Eq 2). The remaining percentage of the total area represents the proportion of green space where human activity is non-detectable and therefore infrequent, if not completely absent. The inverse of this value would be the percent area of the property where human activity is at a non-detectable level throughout the

timeframe.

$$Activity\ Density_{greenspace} = \frac{\sum Activity \times Area_{polygon}}{Area_{greenspace}} \qquad \text{Eq 1}$$

$$Activity\ Coverage_{greenspace} = \frac{\sum Area_{polygon}}{Area_{greenspace}} \qquad \text{Eq 2}$$

To compare activity density and coverage to traditional estimates of visitation and to validate the estimated activity patterns through our adjusted metric, we examined the five Conservation Halton green spaces that had reservation-only access. We summarized the total number of visitors on weekends and weekdays for June, July, and August 2020, with the sum adjusted activity data and fit a linear model. The number of visitors and day-of-week were fitted as interacting predictors.

We compared the density of trails among all green space properties with official trails (16 in total) to activity density and coverage using linear models. We identified any grid cell from the activity data that intersected with the trail network (function *st_intersection*; package *sf*) and summed the area of activity on trails divided by area with human activity. The resulting percentage represents the amount of human activity that is spent on trails. To determine if greater visitation to a green space relates to activity on trails, we fit a linear model comparing the percentage of human activity on trails to the percent of human activity in the property.

Next, we determined how human activity intersected with the Ecological Land Classification Community Class (ELC-CC). The proportion of human activity was determined by dividing the activity coverage in each ELC-CC by the total area of activity coverage. The proportion of ELC-CC used was established by dividing the area of activity coverage in each ELC-CC by the total area of that ELC-CC in the respective green space.

Lastly, we tested if activity density or coverage had any relationship with biodiversity patterns in these green spaces. We compared the basal area of tree species collected as part of a long-term monitoring project at the 11 sites using a partially constrained correspondence analysis (pCCA). We fit the activity density (averaged across weekdays and weekends), activity coverage, and the proportion of weekend activity (i.e., weekend activity divided by weekday activity) as predictors. Year was fit as a conditioning matrix to partial out any interannual differences in tree composition. However, many of these trees were long-lived individuals without substantial differences in species over years. We conducted a permutation test of pCCA (function *anova.cca*, package *vegan*) to determine model significance and percent of variation explained [60].

All analyses were conducted in *R* version 4.1.2 [61]. All scripts and source codes are available on a public repository that can be found at https://github.com/afilazzola/CUERecreationEcology. Data for the characteristics and summarized activity values for each green space tested are also publicly available [62] https://doi.org/10.6084/m9.figshare.21304767.v1.

## Supporting information

**S1 Appendix. Examples of excluding road activity.**
(DOCX)

**S1 Table. Sample Mapbox Movement dataset.**
(DOCX)

**S2 Table. Characteristics of Conservation Halton green spaces.**
(DOCX)

## Acknowledgments

We thank Mapbox for providing the anonymized GPS data of smart devices.

## Author Contributions

**Conceptualization:** Alessandro Filazzola, Kimberly Barrett, Andrea Dunn, Marc T. J. Johnson, James Scott MacIvor.

**Data curation:** Alessandro Filazzola, Garland Xie, Kimberly Barrett, Andrea Dunn.

**Formal analysis:** Alessandro Filazzola, Garland Xie.

**Funding acquisition:** Marc T. J. Johnson, James Scott MacIvor.

**Investigation:** Alessandro Filazzola, Garland Xie, James Scott MacIvor.

**Methodology:** Alessandro Filazzola, Garland Xie, James Scott MacIvor.

**Project administration:** Alessandro Filazzola, Marc T. J. Johnson, James Scott MacIvor.

**Resources:** Alessandro Filazzola, Kimberly Barrett, Andrea Dunn.

**Software:** Alessandro Filazzola.

**Supervision:** Alessandro Filazzola, Marc T. J. Johnson, James Scott MacIvor.

**Validation:** Alessandro Filazzola.

**Visualization:** Alessandro Filazzola, Garland Xie.

**Writing – original draft:** Alessandro Filazzola.

**Writing – review & editing:** Alessandro Filazzola, Garland Xie, Kimberly Barrett, Andrea Dunn, Marc T. J. Johnson, James Scott MacIvor.

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
