## [Decision Letter · Decision Letter 0]

23 Jul 2022

Dear Dr. Filazzola,

Thank you very much for submitting your manuscript "Using mobile cellular data to quantify human activity in green spaces" for consideration at PLOS Computational Biology.

As with all papers reviewed by the journal, your manuscript was reviewed by members of the editorial board and by several independent reviewers. In light of the reviews (below this email), we would like to invite the resubmission of a significantly-revised version that takes into account the reviewers' comments.

We cannot make any decision about publication until we have seen the revised manuscript and your response to the reviewers' comments. Your revised manuscript is also likely to be sent to reviewers for further evaluation.

Sincerely,

Ricardo Martinez-Garcia

Associate Editor

PLOS Computational Biology

James O'Dwyer

Deputy Editor

PLOS Computational Biology

Reviewer's Responses to Questions

**Comments to the Authors:**

Reviewer #1: The authors provide an interesting analysis of human activity in green area, a subject that is is of major importance in the perspective of designing cities that allow for open air activity, especially in pandemic times.

In general the text seems well structured, but it is hard to grasp the meaning of the work carried out by the authors by reading the results and discussion sections.

The interpretation of the results is placed outside of the discussion section, which makes it look like something less important, while actually it is the main key to understand all the results.

The description of the methods is too vague and general, it is never clear if the metrics are measured using a specific month or a temporal average and if seasonality is taken into account.

In the text the authors often refer to their mobility metrics as "mobility data", which is misleading and hinders any understanding of the quantities at play.

Most of the figures only show the results for a sub-set of the total number of green areas present in the dataset (53), but the authors never justify this choice.

I think the authors need to put more effort in the description and justification of the methodology and results in the text, correcting the lexicon and making clear what are the magnitudes measured from the mobility data. For this reason I suggest major revision.

Here a list of major concerns:

- repetition in line 63 wrt 61

- line 115: “By contrast, biodiversity data are rarely collected hourly” it is not clear what type of biodiversity data do the authors expect to collect at a hourly time scale

- Fig2: not clear what is the temporal scale used in the figure, e.g. a day, one week, one month, an average of months?

- line 197: typo “gereen”

- Fig3: plots need correlation scores and pvalues to be shown in the figure

- line 203: “Human activity varied considerably by ELC-CC. Forest and cultural land classes were

204 associated with the most mobility data, followed by talus and cliff” what is the meaning of being associated with the most mobility data? what metric of mobility are you considering here?

- line 207: “By contrast, forests and cultural were used relatively infrequently in proportion to their

abundance among green spaces” is this showed in Fig4B? it was not clear by looking at the picture nor by reading the caption that frequency of visiting was measured by this metric. Indeed I failed to grasp the real difference between Fig.4A and B. Also, again it is not clear what temporal scale are the authors using to measure mobility here.

- Fig5: it is not clear what are the causal hypotheses that are assumed in this analysis. It is interesting to check correlations between different metrics of human activity and trees diversity, however the authors do not explain what are the known hypotheses regarding the influence of human activity on trees diversity in green areas.

- i do not understand why the interpretation of results composes a different section from the discussion. itwas hard to read and grasp the meaning of all analyses and results until a paragraph after discussion finally explained the reasoning behind the hypotheses tested in the results section.

- what is the spatial resolution of biodiversity data? I assume they are aggregated at level of green spaces

- line 421: there are 53 green spaces but only 5 are shown in Fig1 and only 11 in Fig.3, how come the authors discarded the rest of green areas from the analyses?

- the authors often refer to the two mobility metrics showed in equation 1 and 2 as “mobility data”. it really hinders the understanding of the analyses and makes the reasoning of the work much more obscure than it could be. The authors need to rephrase most of the sentenced including “mobility data” to correctly point at the metrics they are referring to when describing the results. Mobility data per se does not correlate, it is the metrics that we extract from them to be quantitatively comparable and measurable. For instance “For every green space, we calculated the total adjusted mobility data” this sentence has no meaning, one thing is to say that mobility data is very informative on human activity, the other is to proper measure some quantities from it. Please rephrase every part in which you refer to mobility metrics, properly using the terms associated with the two metrics that you compute in the Methods section.

- the activity density metric is calculated using the activity provided by Mapbox, this dataset provides the normalized activity in each grid cell of the given green area. Is the activity normalized on the maximum or is it normalized to have the sum of all cells activities = 1? In case it is the second one, this activity in a park with a lot of activity in all the grid cells would provide a very low score.

- the activity density metric has the green area surface at the denominator, does it include all the green area surface or only those cells with a minimum activity required to appear in the dataset?

- In general the authors need to put more effort in the description of the mobility metrics that they design, in order to correctly address the quantitites at play and the temporal scales. For example, the activity metric is the sum of all cells activities divided by the green area total surface, is this sum running over grid cells only or also on time?

Reviewer #2: This manuscript reports the integration of different analytical methods to connect the visitation patterns deduced from mobile phone data to the pathways, land use and species in urban green areas. Although the article is interesting and well-written, I feel that the approach is not too innovative (see for example Le Normand et al, PLOS ONE, 2014). Additionally, the data is not properly described, which difficults the understanding of the quantitative results shown in this manuscript, with only a partial description included after the results. In particular, I suggest the authors to strengthen the following points:

-The mobility is quantitatively introduced in the manuscript, but the used variable is not explained (for example, what is the y-axis in Fig. 1? After log-transforming, I find remarkable that any variable has the value 140). Authors should then explain accurately first what the mobile cell data is, whether call detailed records, high-resolution trajectories..., and secondly this mobility is measured as number of unique users, number of calls, number of points falling on a grid cell, cumulative time spent on a grid cell... There are many options, so I strongly recommend to specify what is measured.

-Fig 1: I recommend using data points, appart from the letters, to know specifically where the point is with respect to the fit. What is the formula of these fits and their results? I find interesting that the slopes are different for weekdays and weekend, and think that some comments on this should be included in the main text.

-L162: this 9.2% higher should be commented. I guess that this is the total mobility after aggregating all the parks. However, this number does not make too much sense for me, because the curves on Fig. 1 are not parallel in a semilog scale. First, I would be happy to see if they are in the log-log scale, and then mentioning a % excess would make sense. Otherwise, it does not make sense, because including parks with higher or lower mobilities would impact this number.

-L177: Fig. 1 is referenced here, but I think this reference should point to Fig. 2.

-Fig. 2: it would be nice to see some text specifying which is Hilton Falls and which is Kelso Conservation Area.

-L203: use of an acronym not previously introduced, ELC-CC.

-Fig. 3: appart from comparing the total activity with the trail density, I suggest comparing the activity on cells as a function of the specific trail density of that cell.

-FIg 3: need to specify the fitted formulas and the results.

Reviewer #3: Summary of the research and overall impression

This paper investigates associations between human occupancy and ecological characteristics of parks in the Greater Toronto Area using a new data resource: aggregated grid-cell mobility data. This case study is an example of one way these data can be applied to understand relationships between park features and patterns in human use. The paper has two areas of contribution: the first is a methodological example for using human mobility data in park management and human-nature research. It also provides a series of findings describing associations between human occupancy and park features, from a sample of 53 parks. This is a novel and compelling contribution to the literature as it is a very early example for relating mobility data to park and ecological features, and probably the first to do so using specifically mapbox data, and other ecological data such as park trails or tree diversity. This paper provides an example for park managers and municipal organisations looking for ways to apply mobility data in management of green amenities. The strength of the paper is in its clarity and systematic study of park usage, and its description of the approach taken for using and preparing the raw mobility data for use. The paper can use some refinement in how it presents related literature, and more clarity in how it structures and portrays the overall contribution.

Evidence and examples

My first and main comments are focused on the Abstract but could relate to the introduction and discussion. These have to do with clarifying the purpose of the paper. It is unclear from the outset if the purpose is to provide methods or results or your models and data. Of course, including both is fine, but the methods-based contribution does not align with your research questions, and does not receive a strong literature component in the discussion or the introduction. I don't see any review of other studies that use mobility data in recent literature. In the abstract you state this is the first time mobility data has been used to quantify human activity in green spaces. This is not the case. Although, it is definitely an early study with important and novel contributions. In addition, you state you provide a framework for mobility data usage. I would soften this term as the term “framework” suggests something more formal than an outline of your own process. Ideally, your approach and recommendations could be highlighted with a summary, table or figure in the discussion as I expect these will be interesting and useful to readers.

In line with my above comments, I agree that your methods will likely be very useful for planners, or park managers, although there was no study conducted with end users of this method nor an analysis of the process compared with other methods. Therefore, one option could be to present the methods as a reflection of your own personal experiences doing the analysis for this study, with some helpful tools (such as R script) and the findings of the study as the core contribution. But the opposite could be the case, where a method is presented (but would include a more systematic methods comparison)

This issue also relates to Table 1, which I believe to be a useful representation of the challenges related to using mobility data and a helpful tool for readers. Although it is not placed within the context of other literature that review these same topics or apply similar data. If the table is in the introduction it should have citations. If you choose to relocate it in the discussion you could frame it specifically around your study and specific data type.

To help here are a few papers that have covered similar topics and could help contextualise your work (as well as observations made in Table 1).

Moro, E., Calacci, D., Dong, X. et al. Mobility patterns are associated with experienced income segregation in large US cities. Nat Commun 12, 4633 (2021). https://doi.org/10.1038/s41467-021-24899-8

Saxon, J. (2021). Empirical measures of park use in american cities, and the demographic biases of spatial models. Geographical Analysis, 53(4), 665-685.

Song, Y., Newman, G., Huang, X., & Ye, X. (2022). Factors influencing long-term city park visitations for mid-sized US cities: A big data study using smartphone user mobility. Sustainable Cities and Society, 80, 103815.

Song, Y., Huang, B., Cai, J., & Chen, B. (2018). Dynamic assessments of population exposure to urban greenspace using multi-source big data. Science of the Total Environment, 634, 1315-1325.

Venter, Z. S., Barton, D. N., Gundersen, V., Figari, H., & Nowell, M. (2020). Urban nature in a time of crisis: Recreational use of green space increases during the COVID-19 outbreak in Oslo, Norway. Environmental research letters, 15(10), 104075.

Yoo, E. H., & Roberts, J. E. (2022). Static home-based versus dynamic mobility-based assessments of exposure to urban green space. Urban Forestry & Urban Greening, 70, 127528.

Korpilo, S., Virtanen, T., & Lehvävirta, S. (2017). Smartphone GPS tracking—Inexpensive and efficient data collection on recreational movement. Landscape and Urban Planning, 157, 608-617.

Kim, J., Thapa, B., Jang, S., & Yang, E. (2018). Seasonal spatial activity patterns of visitors with a mobile exercise application at Seoraksan National Park, South Korea. Sustainability, 10(7), 2263.

Ladle, A., Galpern, P., & Doyle-Baker, P. (2018). Measuring the use of green space with urban resource selection functions: An application using smartphone GPS locations. Landscape and Urban Planning, 179, 107-115.

Vanky, A. P., Verma, S. K., Courtney, T. K., Santi, P., & Ratti, C. (2017). Effect of weather on pedestrian trip count and duration: City-scale evaluations using mobile phone application data. Preventive medicine reports, 8, 30-37.

Rout, A., Nitoslawski, S., Ladle, A., & Galpern, P. (2021). Using smartphone-GPS data to understand pedestrian-scale behavior in urban settings: A review of themes and approaches. Computers, Environment and Urban Systems, 90, 101705.

Related to the above issues, I would also edit your research questions #2. (line 141). I do not believe you cover the question as written. From my readings the question would be better if it related to determining how the anonymized mobility data intersected with the ecological land classification Community Class.

Lines 21-22 Consider the terminology you are using in the title (and possibly) throughout. You start with the term: “mobile cellular data”. I realise this term is appropriate because data is provided as a cell grid - but it could be misinterpreted as data that is aggregated at the cell tower level. Cell tower data has been available for decades and the term may not represent the novel nature of your work. Mapbox provides data that is structured differently than other data providers: where data is aggregated by polygon or point of interest. It is also unique in that it provides a relative rating for visitation rather than raw numbers. I don't know if there is a specific term for this kind of data but I believe the literature should begin to differentiate between mobility data with clear language as it can become confusing, especially for those aiming to compare options. To cover data from mobile phones that infers human travel information the term “mobility data” is often used, although it may not apply to your work as the data does not include movement or travel information, but rather occupation counts for a cell grid. I recommend you review some of this literature and describe your terms and their definitions in the introduction.

It would be helpful to include a description of the landscapes that the study park sample is located in, and citations relating to challenges with management of the parks with these specific ecological or social features in the introduction. Please also indicate earlier in the paper that your sample includes 53 parks, (or 7 for part of the study).

Line 155, please refer here to where in your own document (or elsewhere) that threshold levels are defined by mapbox?

I find FIgure 5 almost impossible to read due to overprinting of text. Is there another visual type that would make these findings more readable (this also applies to figure 3, although less of an issue)

Line 324-325 I think the causation or mechanistic relationship issue would be less of a risk to readers if the sample was more clearly explained earlier - particularly in reference to other parks or landscape types. For example - rock formations are not available in many other areas, and thus, these findings would not be transferable. Although I think a sample of 53 is large enough to provide some generalizability across similar landscapes. Just how general your findings can be would be useful to include here. That said, I think your discussion of the reason for correlations is helpful and clear.

Line 411 One clear advantage of grid-cell mobile data vs data cropped by a polygon boundary is the ability to isolate specific locations near other confounding features. I think the method you use to filter out noise from highways and other non-park uses, is very helpful. The graphic you provide in supplemental materials showing highway locations and nearby cells is useful but I think you should also include more information in the main paper, particularly how much data was removed, the distribution of raw data vs the filtered data and number of removed cells or rows of data due to these changes. This is particularly important if you aim to position your paper as a methods-focused contribution.

Line 481-483 Can you clarify the verbiage here? As a reader I am getting confused about which term refers to the traditional visitation estimates vs visitor counts from the mobility data.

Line 481 - 483, Can you please include models for more advanced linear models, i.e. those that include interactions or multiple variables?

Line 509 - your Github link does not currently open

Please provide summary statistics for raw and cleaned data. Particularly for Mobility data which I believe is not being shared.

**Have the authors made all data and (if applicable) computational code underlying the findings in their manuscript fully available?**

Reviewer #1: **No: **I did not receive any link pointing to the data and/or code used to reproduce the main figures.

Reviewer #2: **No: **The mentioned webpage does not work

Reviewer #3: **No: **The weblink for code sharing does not currently work, although the intention is to share it. I do not believe the authors have made the mobility data available.

PLOS authors have the option to publish the peer review history of their article (what does this mean?). If published, this will include your full peer review and any attached files.

Reviewer #1: No

Reviewer #2: No

Reviewer #3: No
---

## [Decision Letter · Decision Letter 1]

7 Nov 2022

Dear Dr. Filazzola,

Thank you very much for submitting your manuscript "Using smartphone-GPS data to quantify human activity in green spaces" for consideration at PLOS Computational Biology. As with all papers reviewed by the journal, your manuscript was reviewed by members of the editorial board and by several independent reviewers. The reviewers appreciated the attention to an important topic. Based on the reviews, we are likely to accept this manuscript for publication, providing that you modify the manuscript according to the review recommendations.

Sincerely,

Ricardo Martinez-Garcia

Academic Editor

PLOS Computational Biology

James O'Dwyer

Section Editor

PLOS Computational Biology

Reviewer's Responses to Questions

**Comments to the Authors:**

Reviewer #1: The revised version of the manuscript is much clearer than before and the authors answered all my concerns.

I only have few minor comments regarding the description of data and metrics definitions.

A part from this I consider the manuscript suitable for acceptance when these comments will be addressed.

- still not clear what this activity density is, in line 524: “Since the provided Mapbox index represents a normalized proportion, our adjusted metric of activity density relates to the proportion of smart device activity per meter squared.”

Is this smart devices activity computing the number of unique smart devices appearing in those two hours? or single devices may be counted multiple times in the same two hours record?

- line 529 “We totaled the area identified with human activity and divided it by the total area of the property to determine the percent area (Eq. 2). The inverse of this value would be the percent area of the property where human activity is at a non-detectable level throughout the timeframe.”

The first calculation already gives you a percentage, if you take the inverse you do not have a percentage anymore, you need to take the complementary to say what you meant. Maybe the authors meant “the rest of the land… is the percentage of area where human activity is at a non-detectable etc etc”

Reviewer #2: Thanks for addressing the comments, the manuscript has remarkably improved.

Reviewer #3: Thank you for your revisions. I can see that you have made significant changes to the document and have addressed my previous comments.

**Have the authors made all data and (if applicable) computational code underlying the findings in their manuscript fully available?**

Reviewer #1: Yes

Reviewer #2: **No: **They mention that a repository with the code used for the analysis will be public at a later stage.

Reviewer #3: Yes

PLOS authors have the option to publish the peer review history of their article (what does this mean?). If published, this will include your full peer review and any attached files.

Reviewer #1: No

Reviewer #2: No

Reviewer #3: No

Figure Files:

Data Requirements:

Reproducibility:

References:

---

## [Editor Report · Decision Letter 2]

10 Nov 2022

Dear Dr. Filazzola,

We are pleased to inform you that your manuscript 'Using smartphone-GPS data to quantify human activity in green spaces' has been provisionally accepted for publication in PLOS Computational Biology.

Best regards,

Ricardo Martinez-Garcia

Academic Editor

PLOS Computational Biology

James O'Dwyer

Section Editor

PLOS Computational Biology

---

## [Editor Report · Acceptance letter]

18 Nov 2022

PCOMPBIOL-D-22-00584R2 

Using smartphone-GPS data to quantify human activity in green spaces

Dear Dr Filazzola,

I am pleased to inform you that your manuscript has been formally accepted for publication in PLOS Computational Biology. Your manuscript is now with our production department and you will be notified of the publication date in due course.

With kind regards,

Anita Estes
